# Managing Feral Swine: Thoughts of Private Landowners in the West Gulf Region

Nana Tian [1,*], Alyssa Mineau [2] and Jianbang Gan [3]

1   Arkansas Forest Resources Center, College of Forestry, Agriculture & Natural Resources, University of Arkansas at Monticello, Monticello, AR 71656, USA
2   Forest Stewards Guild, 2019 Galisteo St Suite N-7, Santa Fe, NM 87505, USA; am045384@uamont.edu
3   Department of Ecology and Conservation Biology, Texas A&M University, College Station, TX 77843, USA; j-gan@tamu.edu
*   Correspondence: tian@uamont.edu; Tel.: +1-870-460-1849

**Abstract:** Feral swine (*Sus scrofa*) have extensive harmed private landowners throughout the southern United States, especially in the West Gulf Region. Managing feral swine on private land is becoming increasingly critical and challenging to reduce both ecological and economic damage. To better understand private landowners' experience and preferences for various feral swine management measures, we surveyed private landowners across the West Gulf Region (WGR) including Arkansas (AR), Louisiana (LA), and East Texas (ETX) in 2021. A total of 4500 surveys were mailed across the three states, with 2000 questionnaires sent in AR, 1500 in LA, and 1000 in ETX. Using descriptive analysis and principal components analysis (PCA), we analyzed private landowners' experiences and preferences in feral swine management across this region. The tesults revealed that most private landowners (>85%) in the region were familiar with feral swine, and over 80% of them had ever seen the intrusion of feral swine onto their properties. Regarding the potential management measures, these landowners strongly supported lethal control methods such as capture and kill; in addition, they expressed a strong desire to receive education on and technical assistance with controlling feral swine. These findings provide a better understanding of private landowners' familiarity and experiences with feral swine presence on their properties and their preference and support for various feral swine control measures, aiding in developing more effective feral swine management and control policies and programs in the West Gulf Region and beyond.

**Keywords:** feral swine; West Gulf Region; private landowners; control measures





## 1. Introduction

Feral swine (*Sus scrofa*) are an invasive species in the United States (US), introduced in the 1500s by Spanish colonizers who traveled and settled down throughout the southern United States with their domestic pigs (*Sus scrofa domestica*) [1]. In the 1800s, Eurasian wild boars (*Sus scrofa*) were introduced into further northern regions of the US for hunting. The longstanding southern populations of domestic pigs interbred with the newly introduced northern wild boar populations, resulting in hybrid offspring known today as "feral hogs", "feral swine", "wild pigs", and other variations [2]. Since their introduction, these feral swine populations have rapidly spread across the US, with estimates of there being around seven million in the country [3], with nearly half of them concentrated in the West Gulf Region (WGR), including Arkansas, Louisiana, and Texas. The presence of feral swine imposes numerous challenges for private landowners and land managers, causing extensive damage in the areas they inhabit.

Due to their early maturation and high reproductive rate [4,5], feral swine groups, known as "sounders", can rapidly increase in size, sometimes comprising over 50 individuals in a sounder. This growth can lead to widespread destruction across the landscape, impacting various stakeholders in the WGR. For land managers, the foraging behaviors

of feral swine, known as "rooting", can disrupt normal soil chemistry, mix soil horizons, and subsequently alter local vegetative communities, contributing to the spread of other invasive species [6]. On a larger scale, this soil disturbance may worsen soil erosion and cause damage to sensitive ecological areas and critical habitats for species of concern, particularly within wetlands [7]. The southeastern US, which includes the WGR, harbors 43% of the nation's palustrine and estuarine wetlands [8], which are facing severe impacts from the presence of feral swine. Both within and beyond the WGR, feral swine prey on numerous vulnerable and endangered species in these wetlands, such as the American alligator (*Alligator mississippiensis*), loggerhead turtles (*Caretta caretta*), and reticulated flatwoods salamanders (*Ambystoma bishop*) [9–11].

Feral swine, as an invasive species, are a worldwide issue. For example, according to Risch et al. [12], feral swine are identified as one of the "100 of the World's Worst Invasive Alien Species" given that this species is both a large predator and herbivore throughout their native and non-native range [12]. In addition to their direct impact on wildlife and plants, they can disturb ecosystem structures through rooting and digging behavior [12,13]. Given all the damage that feral swine can cause to different natural resources and ecosystems, they are becoming a worldwide issue. For example, Clarissa Alves et al. [14] reported the feral swine problems in Brazil and found that the lethal control method of hunting with dogs was the main technique used for controlling feral swine. Bengsen et al. [15] summarized the feral swine issues in Australia and New Zealand, and projected their further expansion and distribution in both countries and damage to the environment, agriculture, and natural resources. Massei et al. [16] reviewed possible methods, including lethal (i.e., shooting, trapping) and nonlethal (i.e., fencing, translocation, etc.) methods, to mitigate the impact of feral swine and concluded that combining different control methods and establishing posteradication monitoring to ascertain the eradication succeeded for the island area.

For private landowners, feral swine add significant threats to forestlands, croplands, and pasturelands [13]. For example, the average economic loss due to feral swine damage was estimated at 67.13 USD/ha, 42.96 USD/ha, 27.31 USD/ha, and 57.54 USD/ha for landowners in the region who owned cropland, forestland, pastureland, and multiple land types, respectively [13]. In hardwood forests, a key component of the feral swine diet is acorns [17], and their presence leads to a reduction in acorn availability and limits forest regeneration [18]. Moreover, feral swine foraging can result in forest regeneration failure by rooting seedlings and disturbing the roots of recently planted pines and hardwoods [19]. In agricultural lands where feral swine are present, crop yields can decrease [20–22], and complete decimation of crop fields is not uncommon [23]. In pasturelands, the risks stem from feral swine interactions with livestock. Rooting by feral swine in pasturelands can encourage the growth of undesirable grass species, altering the plant composition in those areas [24] and affecting overall livestock health due to reduced food availability. Additionally, feral swine can transmit diseases to domestic livestock, including swine brucellosis, pseudorabies, classic swine fever, and African swine fever [25]. These diseases are known to cause birth defects and/or death in both livestock and wildlife species [26]. While these diseases have been eradicated from the pork industry in the US, feral swine could serve as reservoirs, potentially reintroducing them to domestic pigs and causing significant losses to the US pork industry [27].

To mitigate the damage and negative effects caused by feral swine, effective management/control is essential. Currently, feral swine are primarily controlled through both lethal and nonlethal measures. Common lethal controls include shooting/hunting (either aerial or ground-based), trapping and euthanizing, and poisoning. Common nonlethal controls involve trapping and relocating, installing fences, harassing (e.g., noise, lights), and using repellents (e.g., scents, pepper spray). Previous studies have demonstrated varying levels of success with these control measures. Lethal methods like shooting are cost-effective in reducing the feral swine population [28], but some researchers argue that it fosters a "hunting culture" that increases tolerance of feral swine presence [29] or may encourage illegal transport for recreational purposes into areas where feral swine did not

previously inhabit [30]. Moreover, shooting is often perceived as a "short-term solution" since hunting has not proven to significantly reduce the local feral swine populations [31]. Methods like trapping are popular due to their speed, reusability, and ability to capture a large group at once, but some studies (e.g., [16,25]) indicate a loss of efficacy in regions with abundant food availability or in sounders that have developed trap shyness. Additionally, trap utilization tends to be labor-intensive, potentially discouraging frequent usage by some landowners.

In the West Gulf Region, the current feral swine population is reported as 2.4 million in Texas [32], 700,000 in Louisiana [33], and 200,000 in Arkansas [34]. Managing and controlling feral swine, particularly on private lands, are of growing importance, given that most lands in this region are privately owned [35]. Therefore, the specific objectives of this study were to (1) enhance the understanding of private landowners' experience in feral swine management in the West Gulf Region and (2) examine their preference and support for various feral swine control measures. The findings of this study foster a better understanding of private landowners' experiences with feral swine and their preference and support for various management and control measures. Such knowledge is invaluable to wildlife management personnel, natural resource managers, and policymakers in the study region and other areas faced with similar feral swine problems. A nuanced understanding of the perspectives of landowners fosters information sharing between landowners and land managers as well as management participation/compliance, which are essential for developing comprehensive, effective, and sustainable strategies to manage feral swine and other invasive species.

## 2. Methods

### 2.1. Study Area

To gain insights into private landowners' experiences and support for different feral swine management and control measures, we conducted a mail survey in 2021 across the WGR, encompassing Louisiana (LA) parishes (64), Arkansas (AR, 65), and east Texas (ETX, 38). The surveyed area was mapped by Mineau et al.; see Figure 1 in [36]. The inclusion of east Texas was specific to its geographical proximity to AR and LA, along with shared characteristics such as vegetation cover types, land use, and climatic conditions, all of which are relevant factors in feral swine presence/population dynamics and their management. The tristate area comprising Arkansas, Louisiana, and Texas falls within a humid subtropical climate, with occasional incursions of cold air during winter. The predominant land use types in this region include agriculture (such as rice and crops), forestland, pastureland, ranches, etc. A significant portion of the land in these states is privately owned. Taking forestland as an example, there are approximately 821,000 family landowners (nonindustrial) who collectively own 32 million acres of forests across Arkansas, Louisiana, and Texas [36]. In addition, with the current feral swine population of 2.4 million in Texas [32], 700,000 in Louisiana [33], and 200,000 in Arkansas [34], managing feral swine is becoming increasingly important. This three-state study enables us to better understand private landowners about their experiences and support for managing and controlling feral swine in this border region/landscape.

### 2.2. Survey Instrument

The survey instrument was developed following the Dillman Tailored Design Method [37], and the University of Arkansas at Monticello's Institutional Review Board (IRB# FNRf-01) reviewed and approved the instrument and protocol. Before designing the survey questions, we consulted with wildlife extension experts in the University of Arkansas System, the Division of Agriculture Cooperative Extension Service, to better understand the feral swine issues in the region. Then, we pretested the questionnaire by distributing it to multiple experts who had worked closely with private landowners in this area. Following iterative rounds of refinement and feedback, we finalized the questionnaire before dispatching the survey package.

The cover page provided landowners with assurance regarding the confidentiality of their responses and emphasized that their participation was voluntary. This was followed by a 10-page questionnaire and an electronic consent letter. We obtained mailing addresses from Dynata Inc., targeting private landowners with at least 30 acres (12.14 ha) of land. A total of 4500 surveys were mailed across the three states, with 2000 questionnaires sent in Arkansas, 1500 in Louisiana, and 1000 in east Texas. A reminder postcard was sent two weeks after mailing out the survey package. In this study, we focused on the questions of landowners' support/agreement levels for different feral swine management and control measures. All the options included in the survey were summarized after a thorough literature review. The survey questions included Likert scale items gauging private landowners' agreement or support levels (1 = strongly agree/support, 5 = strongly disagree/oppose) on various feral swine statements and control measures. Additionally, other questions in the survey were categorized into three groups: sociodemographics (i.e., age, gender, education, etc.), ownership characteristics (i.e., acreage, tenure, etc.), and experience (i.e., familiarity) with feral swine.

We analyzed the survey data using descriptive statistics and principal components analysis (PCA). All statistical results are reported at a significance level of 5% ($\alpha = 0.05$). The PCA was used to group the various feral swine management options into different categories and to examine the commonality among different categories.

## 3. Results

### 3.1. West Gulf Region Respondents

Upon receiving the responses, we excluded 285 surveys from Arkansas, 175 from Louisiana, and 86 from east Texas as they were deemed ineligible (undeliverable addresses, death, etc.). The number of usable returned questionnaires was 361 in Arkansas, 319 in Louisiana, and 226 in east Texas, resulting in a response rate of 21.05% (AR), 24.08% (LA), and 24.73% (ETX), with an adjusted response rate of 22.6% for the WGR. Across the WGR, 62.2% of the respondents were 65 years or older, and 84.6% were men. Their education levels varied, with most of the respondents reporting having some level of college or more (73.5%) and 26.5% having a high school education or less. The WGR respondents' income levels also varied, with the majority (78.2%) earning USD 50,000 or more, and 21.8% earning <USD 20,000–49,999. The mean tenure of ownership was reported as 35.1 years (SD = 22.6 years), and the median ownership size was estimated at 323 acres (131.12 ha) (Table 1). A nonresponse bias check was not conducted due to time and resource limitations. However, we found similarities in several key demographic characteristics including age and gender between the sample and surveys of the surrounding regions [38–40].

**Table 1.** Descriptive statistics of sociodemographic characteristics of survey respondents in the West Gulf Region.

| Variable | | Number or Frequency |
|---|---|---|
| Age | *n* (number of valid responses) | 797 |
| 34 years or younger | | 1.4% |
| Between 35 and 44 | | 4.9% |
| Between 45 and 54 | | 7.7% |
| Between 55 and 64 | | 23.8% |
| 65 years or older | | 62.2% |
| Gender | *n* | 800 |
| Male | | 84.6% |
| Female | | 12.9% |
| Prefer not to answer | | 2.3% |
| Other | | 0.3% |

**Table 1.** *Cont.*

| Variable | | Number or Frequency |
|---|---|---|
| Education | *n* | 788 |
| Less than high school/GED | | 3.0% |
| High school/GED | | 23.5% |
| Some college | | 25.4% |
| Associate degree | | 4.9% |
| Bachelor's degree | | 27.7% |
| Advanced degree | | 15.5% |
| Income (USD) | *n* | 692 |
| Less than 20,000 | | 2.6% |
| 20,000–49,999 | | 19.2% |
| 50,000–79,999 | | 27.5% |
| 80,000–100,000 | | 13.0% |
| More than 100,000 | | 37.7% |
| Land ownership characteristics | | |
| Tenure (years of ownership) | *n* | 839 |
| Mean | | 31.5 |
| Acreage owned | *n* | 769 |
| Median | | 323 |

### 3.2. Private Landowners' Experience with Feral Swine

We explored several facets of private landowners' experiences with feral swine, including their familiarity with feral swine, the presence of feral swine on their property, the extent of feral swine presence, and their perspectives on how the feral swine population might change based on their experiences. When asked about their familiarity, 87% of ETX respondents indicated being very familiar with feral swine, while it was 52% in LA and 46% in AR. Only a small percentage of respondents reported not being familiar at all with feral swine across the WGR:13% in AR, 9% in LA, and less than 1% in ETX (Table 2).

**Table 2.** Descriptive statistics of respondents' experience with feral swine in Arkansas (AR), Louisiana (LA), and east Texas (ETX).

| Question Statement | Number or Frequency | | |
|---|---|---|---|
| | AR | LA | ETX |
| Before receiving this survey, which of the following most accurately describes your familiarity with feral hogs? | *n* = 343 | *n* = 246 | *n* = 215 |
| Not familiar at all | 12.8% | 9.0% | 0.5% |
| Somewhat familiar | 41.7% | 38.7% | 12.6% |
| Very familiar | 45.5% | 52.3% | 87.0% |
| Based on your experience, what do you think the population of feral swine will change in your area in the next 5 years? | *n* = 331 | *n* = 277 | *n* = 219 |
| Increase | 62.2% | 75.1% | 90.4% |
| Will not change | 10.3% | 6.5% | 0.5% |
| Decrease | 1.8% | 0.4% | 5.0% |

When asked about whether they had ever seen any feral swine or signs of feral swine on their property, most of the respondents (over 80%) reported Yes. Therefore, a follow-up question was asked to identify the extent to which they noticed feral swine on their land, which included four options: "all over my land (i.e., more than 75% of my land)", "about

half of my land (i.e., about 50% of my land)", "about 25% of my land (i.e., about 25% of my land)", and "a very small portion of my land (i.e., less than 5% of my land)". The descriptive statistics are summarized in Figure 1. Over half of the respondents (57.3%) from ETX reported that they noticed feral swine or the signs of feral swine on all of their land, whereas this percentage was 26.6% in LA and only about one-fifth (20.3%) in AR. By contrast, over half (50.6%) of the respondents from AR reported a very small portion of their land with signs of feral swine presence, being 42.5% in LA and 9% in ETX.

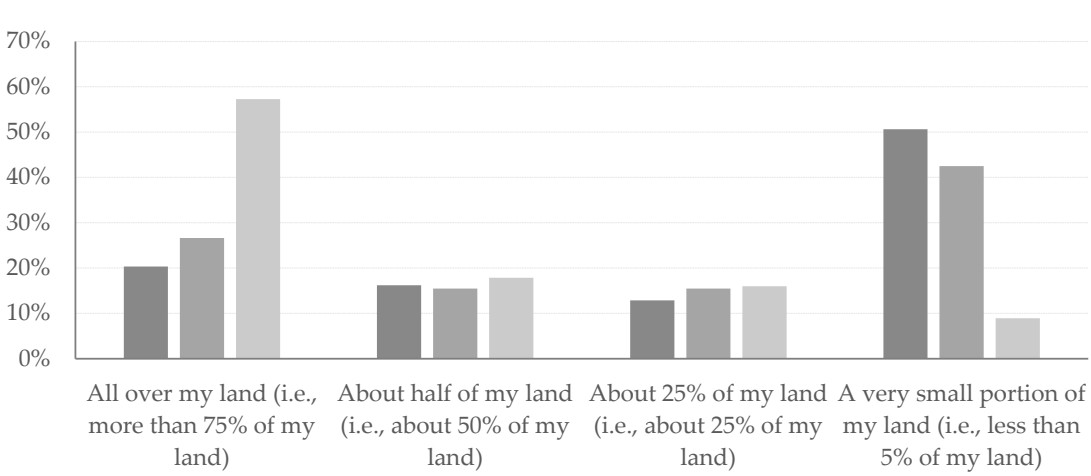

**Figure 1.** The extent of feral swine presence on private lands in Arkansas (AR), Louisiana (LA), and east Texas (ETX).

We then asked how they perceived the feral swine population might change (increase, decrease, or not change) based on their experiences, and most of them anticipated that the population would increase. Specifically, over 90% of respondents in ETX reported an increasing trend in the feral swine population, and this percentage was 75% in LA and 62% in AR (Table 2).

### 3.3. Private Landowners' Support for Controlling Measures

To gauge private landowners' preferences toward different management and control measures for feral swine across the WGR, respondents were prompted to express their level of support on a scale ranging from on (strongly support) to five (strongly oppose) for each potential method. As shown in Table 3 and the figures included in the Appendix A (Figures A1–A3), the average support was high for "Capture and kill" (1.35), "Provide technical assistance for landowners/farmers to control feral swine" (1.55), and "Educate people on how to prevent damage" (1.73), "Targeted sharpshooting on the ground over bait sites" (1.77), as well as "Aerial control using helicopters" (1.88). For example, over 82% of respondents strongly supported the lethal measure of "Capture and kill" to control feral swine, indicating that this lethal management method was widely acceptable among private landowners. By contrast, respondents indicated strongly opposed to "Do nothing (4.71)" and "Capture and relocate (4.11)". For instance, 82% of respondents reported strongly opposing "Do nothing", suggesting private landowners' high desire and willingness to do something to control feral swine.

In addition, to identify prevalent themes in acceptability across the WGR, we conducted PCA on the 11 possible management and control actions, and the results are summarized in Table 4. When performing PCA, orthogonal varimax rotation was applied to create factors without intercorrelated variables, and the coefficients of Cronbach's alpha were estimated to identify the internal consistency within indices. Based on the acceptable Cronbach's alpha score, PCA yielded three factors, which are displayed in Table 4. According to the items included in each factor, we defined PC1 as Nothing and Relocation, PC2 as

Lethal Control; and PC3 as Education- and Incentive-Based; In combination, those three factors accounted for 50% of the total variance.

**Table 3.** Summary of respondents' acceptability of various feral swine control measures.

| Feral Swine Control/Manage Options | Strongly Support | Somewhat Support | Neutral | Somewhat Oppose | Strongly Oppose |
|---|---|---|---|---|---|
| Do not do anything (*n* = 727, mean = 4.71) | 2.2% | 2.1% | 7.6% | 6.2% | 82.0% |
| Capture and remove using trained dogs (*n* = 737, mean = 2.32) | 36.9% | 23.1% | 23.4% | 7.8% | 10.2% |
| Targeted sharpshooting on the ground over bait sites (*n* = 739, mean = 1.77) | 55.8% | 24.3% | 14.6% | 2.6% | 4.3% |
| Capture and relocate (*n* = 727, mean = 4.11) | 7.4% | 9.2% | 12.0% | 8.1% | 63.3% |
| Capture and kill (*n* = 762, mean = 1.35) | 82.5% | 12.9% | 6.5% | 1.1% | 1.8% |
| Aerial control using helicopters (*n* = 740, mean = 1.88) | 55.4% | 18.6% | 19.0% | 2.9% | 5.9% |
| Use legal toxicants (*n* = 740, mean = 2.35) | 45.7% | 14.9% | 17.9% | 6.5% | 16.9% |
| Allow sales of feral hogs (*n* = 735, mean = 2.94) | 28.7% | 15.5% | 21.2% | 4.7% | 30.9% |
| Educate people on how to prevent damage (*n* = 738, mean = 1.73) | 56.0% | 24.5% | 16.5% | 1.4% | 3.2% |
| Provide technical assistance for landowners/farmers to control feral swine (*n* = 742, mean = 1.55) | 65.7% | 24.3% | 9.5% | 1.1% | 1.4% |
| Offer money rewards (bounties) to whoever controlling feral swine (*n* = 743, mean = 1.89) | 50.3% | 22.8% | 22.4% | 2.8% | 3.9% |

*n* denotes the number of valid responses received.

**Table 4.** The PCA results of respondents' support for feral swine control methods.

| | Management Option | Mean | Factor Loading | Eigenvalues | Cronbach's Alpha |
|---|---|---|---|---|---|
| Nothing and Relocation | Do not do anything | 4.73 | 0.332 | 1.495 | 0.641 |
| | Capture and relocate | 4.13 | 0.567 | | |
| Lethal Control | Capture and remove using trained dogs | 2.34 | 0.67 | 1.165 | 0.699 |
| | Targeted sharpshooting on the ground over bait sites | 1.78 | 0.684 | | |
| | Capture and kill | 1.35 | 0.617 | | |
| | Aerial control using helicopters | 1.87 | 0.555 | | |
| | Use legal toxicants (e.g., warfarin, sodium nitrite) | 2.34 | 0.412 | | |
| Education- and Incentive-Based | Educate people on how to prevent damage | 1.74 | 0.348 | 3.236 | 0.661 |
| | Provide technical assistance for landowners/farmers to control feral swine | 1.78 | 0.582 | | |
| | Offer money rewards (bounties) to whoever controlling feral swine | 1.90 | 0.557 | | |
| | Provide subsidies (compensations) to landowners/farmers for feral swine damage | 2.07 | 0.518 | | |

## 4. Discussion

Feral swine, as a highly destructive invasive species, have extensively spread across the United States. Given their rapid population growth and consequential damage to various ecosystems, it is imperative to manage feral swine to safeguard wetlands, forest-lands, agricultural lands, livestock, infrastructure, and more [41]. Managing feral swine cannot be solely accomplished by land/natural resource managers or on public lands alone: private landowners play a significant role given the size of the land they own collectively, especially in the southern US. Therefore, it is crucial to examine what private landowners think about feral swine control and their support and the acceptability of management/control measures.

The findings indicate a notable level of awareness among private landowners in the WGR regarding the presence of feral swine on their properties. A significant majority of these landowners were not only familiar with feral swine but also acknowledged the active presence of feral swine on their land. The results indicated that private landowners in ETX found greater ranges and wider distribution of feral swine than in the other two states of AR and LA. This could be attributable to a bigger feral swine population [32] and the early feral swine history in Texas. Specifically, feral wine were imported and introduced to Texas by ranchers and sportsmen for sport hunting in the 1930s [17,24,28]. The knowledge of the feral swine prevalence among private landholdings underscores the importance of acquiring detailed information on the abundance of this invasive species across the WGR. Gaining insights into the extent of feral swine infestation in this region is crucial for informed decision making by resource managers. These data serve as a key indicator of the urgency associated with providing timely and targeted technical assistance for feral swine control. By understanding the scope of the issue, managers can tailor their interventions to effectively address the specific challenges posed by feral swine in the WGR. This proactive approach ensures that resources are allocated strategically, maximizing the impact of control measures and fostering a more resilient landscape.

Furthermore, this study sheds light on the shared perspectives of private landowners regarding their endorsement of various feral swine control measures across the WGR. Notably, a substantial majority of private landowners expressed strong support for measures such as "Capture and Kill" and "Targeted sharpshooting on the ground over bait sites". This may be due to the fact that lethal control methods such as capture and kill and shooting can be effective, especially when single or small groups of feral swine are found on the property, which is also the method most commonly used by landowners [16,25]. Additionally, there is notable enthusiasm for educational- and incentive-based initiatives, as evidenced by the positive reception of statements like "Educate people on how to prevent damage" and "Provide technical assistance for landowners/farmers to control feral swine". These findings resonate with the attitudes observed among Tennessee landowners, as reported in [42], where respondents similarly favored capture and kill, targeted sharpshooting, education, and the provision of technical assistance. The consistency in attitudes toward these management options not only reaffirms the prevailing sentiments within the WGR but also underscores the broader regional alignment in strategies favored by private landowners. The robust support for these specific management options provides a compelling rationale for the development and implementation of more appropriate feral swine management plans at both the state and federal levels. Recognizing the resonance of these measures with private landowners, such plans can be designed to be not only effective but also aligned with the preferences and priorities of stakeholders. This alignment ensures that management protocols are not only robust in addressing feral swine issues but are also cost-effective, accessible, and enjoy broad-based support, thereby enhancing their overall efficacy in mitigating the impact of feral swine across the region.

Additionally, the findings underscore a resolute stance among private landowners in the WGR against adopting a passive approach, as evidenced by their strong opposition to the option of "Do nothing" when it comes to addressing feral swine issues. This inclination reveals a clear desire and expectation among private landowners to take proactive measures

in controlling feral swine on their properties. Conversely, the method of "capture and relocate" encountered robust resistance from private landowners in this region, aligning with the conclusions drawn by [42] in their study of Tennessee landowners. The documented low support for capturing and relocating feral swine can be attributed to the recognized risks associated with translocation. This sentiment aligns with the concerns highlighted by [24], where translocating feral swine was identified as a potential vector for spreading diseases into new areas. The perceived risk of introducing diseases to regions that were previously unaffected raises substantial apprehensions among private landowners. This concern is particularly relevant to livestock producers and others whose livelihoods are directly impacted by diseases. Consequently, the aversion to the "capture and relocate" method is grounded in a collective understanding of the potential adverse consequences associated with disease transmission, reinforcing the need for careful consideration of the ecological implications and associated risks when devising feral swine management strategies in the WGR.

The results of this study reveal a notable diversity in the levels of support among private landowners for various feral swine control options, aligning with the observations made by [41], who identified significant variations in people's perspectives on management actions. The complexity of feral swine control and management becomes evident as no single measure universally satisfies the objectives of all stakeholders. Consequently, private landowners and natural resource managers employ a range of tools and strategies to control feral swine populations effectively. These nuanced findings hold valuable implications for wildlife management personnel and feral swine managers operating in the WGR. Recognizing the spectrum of private landowners' perspectives on different management options is crucial for assessing the feasibility and acceptability of various approaches. The data obtained from this study can inform the development of strategies with a high likelihood of acceptability, a key factor for successful implementation. The insights garnered from this research not only contribute to the refinement of feral swine management plans but also serve as a valuable resource for wildlife managers and agencies engaged in outreach and education services for private landowners. Crafting strategies that align with the diverse but differentiated preferences identified in this study is integral to fostering successful collaborations among stakeholders and ensuring the effectiveness of feral swine control initiatives. Beyond its practical applications, this study adds depth to the existing literature on the human dimensions of feral swine management and control, thereby advancing our understanding of the complex interplay between stakeholders and management strategies.

## 5. Conclusions

This research delved into the experiences of private landowners grappling with the presence of feral swine across the expansive West Gulf Region (WGR) and explored their endorsement of various control and management measures. This study revealed that private landowners within the WGR were actively contending with the challenges posed by feral swine on their properties and shared a commonality in their support for diverse controlling actions.

Specifically, a significant majority of private landowners expressed favorability toward the implementation of the lethal measure "Capture and Kill" for controlling feral swine, highlighting their proactive stance in addressing this issue. Conversely, there is a collective opposition to the passive approach of "Do nothing", indicating a prevailing willingness and expectation among private landowners to undertake tangible actions for feral swine control. To deepen our understanding, further investigation into the extent of efforts private landowners have invested in managing feral swine on their lands is warranted. Additionally, exploring factors such as economic status and damage loss could unveil influential determinants shaping private landowners' decisions regarding feral swine control actions.

Furthermore, this study underscores private landowners' robust support for receiving education and technical assistance in feral swine management and control. This enthusiastic response signals a clear need for expanded outreach and education programs tailored to the specific needs and preferences of private landowners. As we navigate the complexities of feral swine management, these findings provide valuable insights that can guide the development of targeted initiatives, fostering a collaborative approach between stakeholders and ensuring the efficacy of control measures within the West Gulf Region. The findings of this study can help wildlife management personnel and natural resource managers better understand private landowners' experiences with feral swine and their preference and support for various management and control measures. These results can also aid policymakers in developing more effective feral swine management and control policies and programs in the West Gulf Region and beyond.

**Author Contributions:** Conceptualization, N.T. and A.M.; methodology, N.T., A.M. and J.G.; formal analysis, A.M. and N.T.; writing—original draft preparation, N.T. and A.M.; writing—review and editing, J.G.; project administration, N.T. All authors have read and agreed to the published version of the manuscript.

**Funding:** This study was financially supported by the Arkansas Forest Resources Center, part of the University of Arkansas System Division of Agriculture.

**Data Availability Statement:** The raw/original data are not publicly available due to containing information that could compromise the privacy of research participants.

**Acknowledgments:** We are thankful to the respondents for their time and effort in completing the survey and for the support provided by the Arkansas Forest Resources Center, College of Forestry, Agriculture & Natural Resources in completing this study.

**Conflicts of Interest:** The authors declare no conflicts of interest.

**Appendix A**

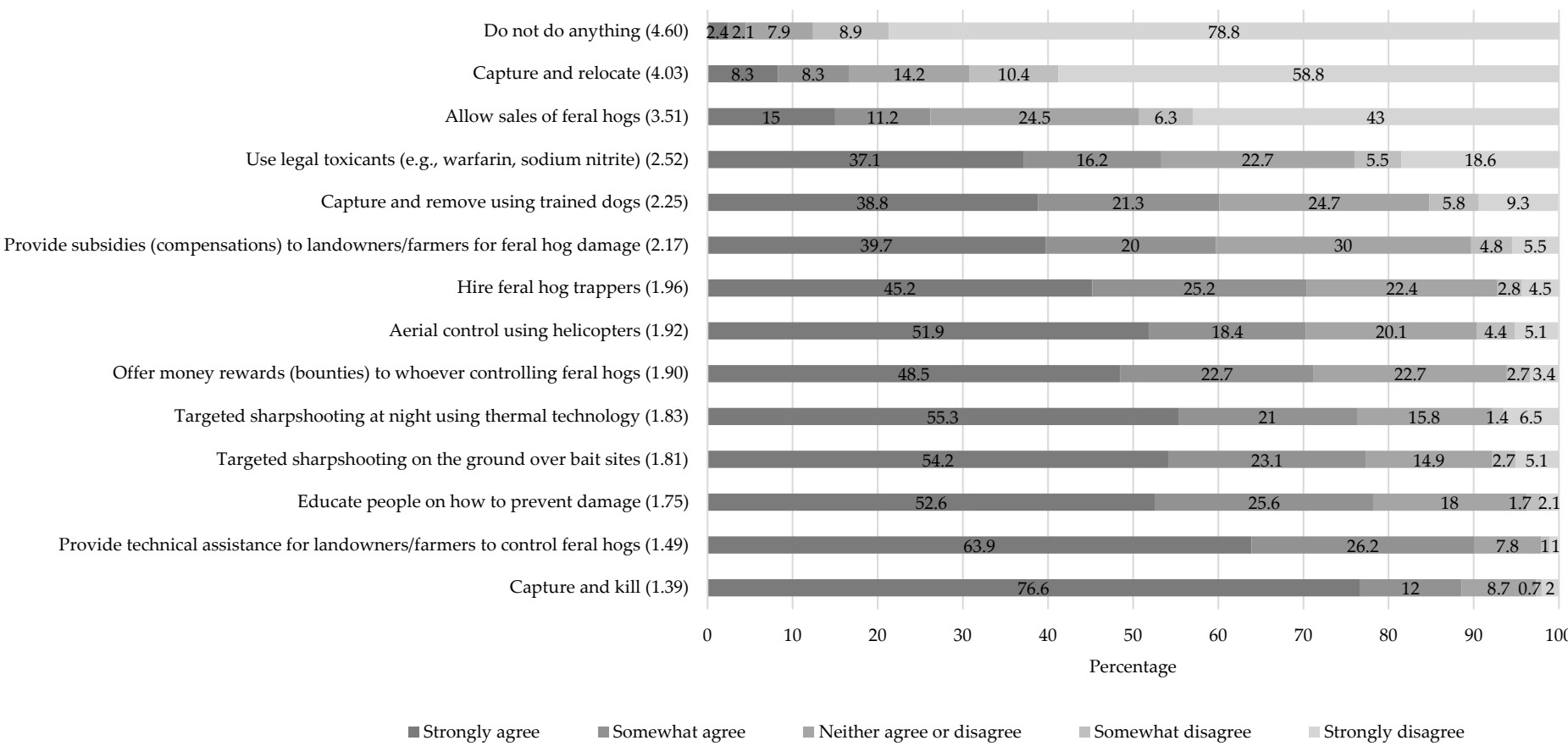

**Figure A1.** Descriptive statistics of AR respondents for acceptability of feral swine control measures.

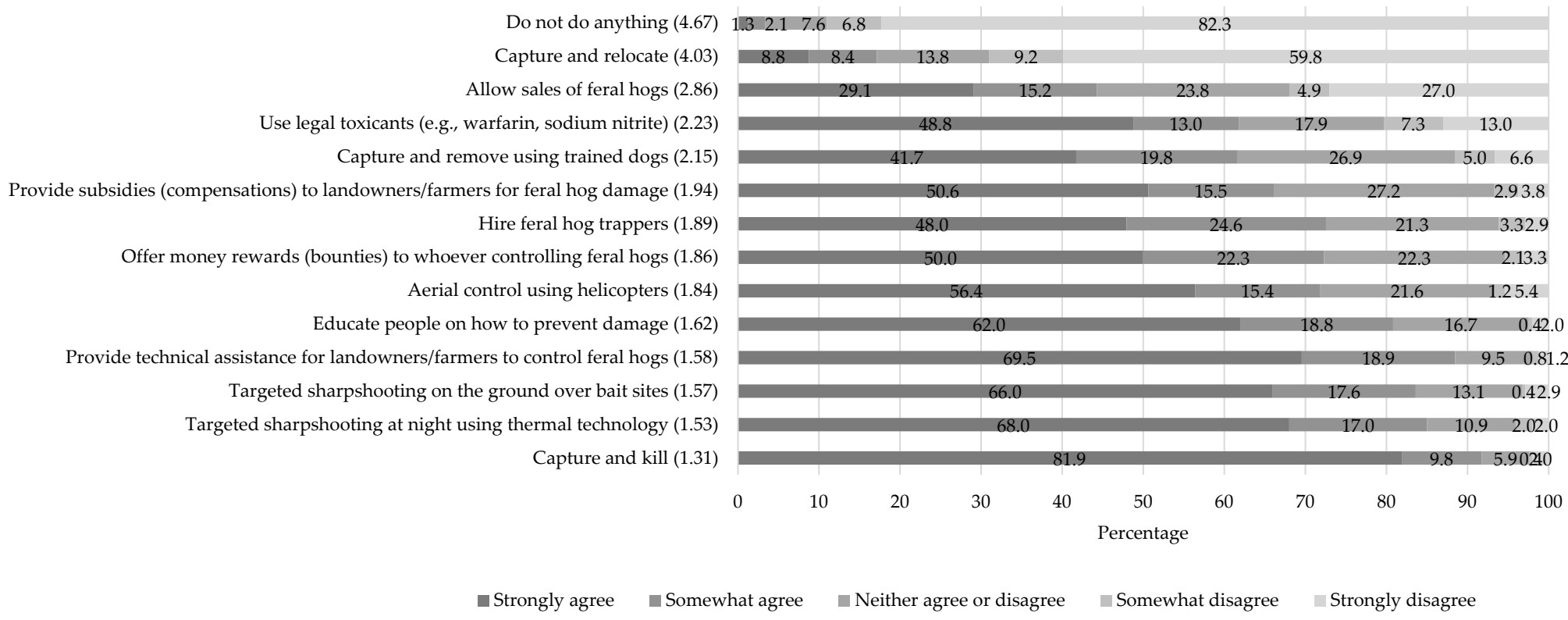

**Figure A2.** Descriptive statistics of LA respondents for acceptability of feral swine control measures.

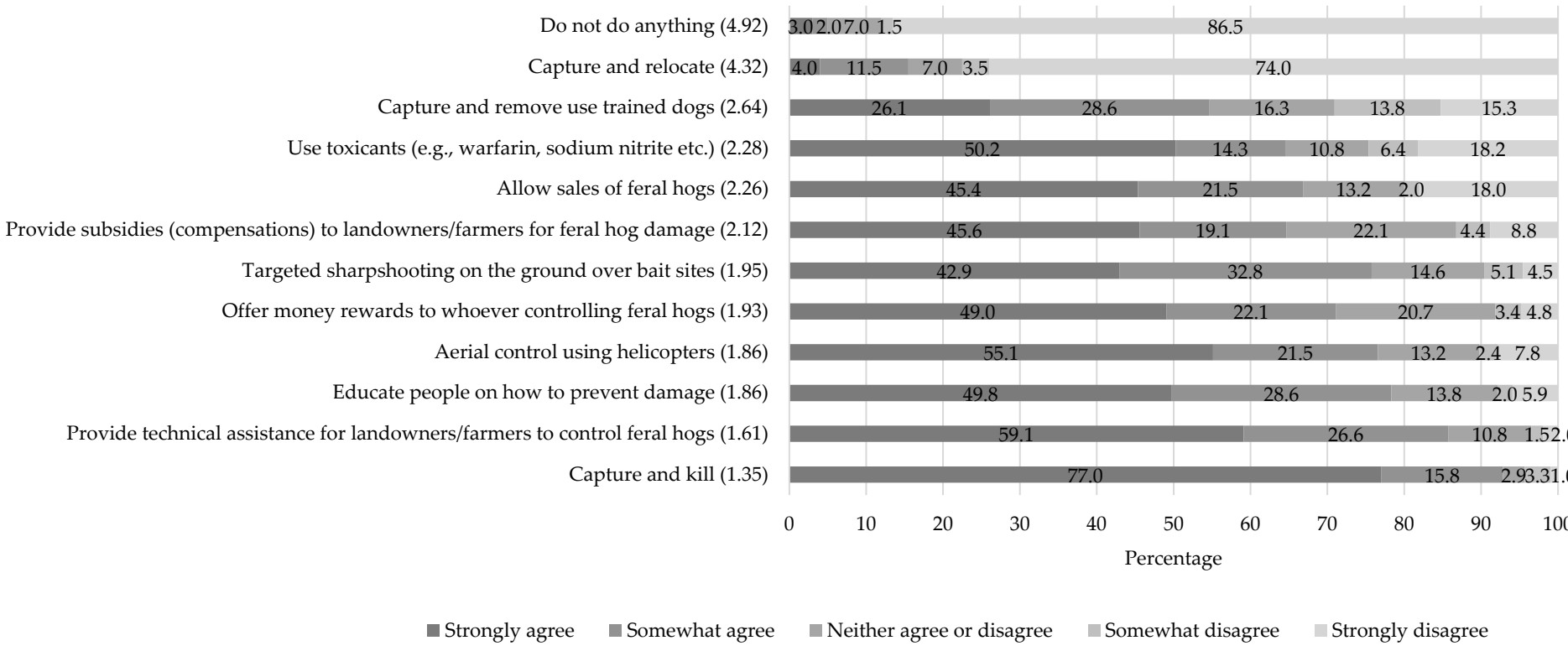

**Figure A3.** Descriptive statistics of ETX respondents for acceptability of feral swine control measures.

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
