# Peer review of "Managing Feral Swine: Thoughts of Private Landowners in the West Gulf Region"

_forests, doi:10.3390/f15030436_

Round 1
Reviewer 1 Report
Comments and Suggestions for Authors
The manuscript is focused on the attitudes toward feral swine among private landowners. The text is divided into the chapters typical for quantitative approach. Authors used adequate statistical methods and it is written in understandable form in scientific form. The Text is written on high level, mainly from results till the end of the manuscript. The comments are presented below.
1. Please use passive in the whole text, it is typical for the scientific papers.
2. The theoretical part is focused on the zoological and ecological level of feral swine in United States and also on the historical migration of feral swine in this region, but this part should be smaller. The Introduction part is relatively missing, please highlight the importance of this study.
3. The kinds of information about psychological aspects of the concept “attitudes” is missing. Please add it into the theoretical background. And also add the current research toward this problematic, not only from US, but all over the world.
4. The research aims should be presented after theoretical background and they are missing, so please add them.
5. The methodological chapter is too brief and it should be divided into the subchapters typical for psychological research. Please read some studies regarding to the field of study and revise this chapter according them.
6. The discussion chapter should be focused on the research aims, which are not presented,
I hope my comments are helpful
Author Response
Please see the attached letter, thanks!

Reviewer 2 Report
Comments and Suggestions for Authors
The study conducted an extensive questionnaire survey of three regions on the west coast of the United States to learn about private landowners' experiences with feral hogs and preferences for various feral hog management measures, providing an important data base for the subsequent development of measures to manage feral hogs. However, the article may be slightly deficient in experimental design, the questionnaire survey is relatively simple and does not investigate the conflict between private landowners and wild boar in more depth, and it is recommended that the authors make some modifications to the experimental design. The modifications of the article as a whole are as follows:
Abstract:
Specific information about the survey methodology is missing from the abstract, and it is recommended that the authors briefly describe the survey design, sample size, and other background information. In addition, it is recommended that the authors conclude the abstract with a brief description of the main significance and contribution of the study.
Introduction:
In the first three paragraphs of the introduction, although the author introduces some of the negative impacts that the presence of wild boars may bring, the author does not introduce specific research cases, the introduction of specific case studies can enhance the persuasive power of the argument, and it is recommended that the author search the literature extensively to introduce some relevant research cases.
Method and Results:
1. The text only briefly mentions that the survey program was developed according to Dillman's customized design method. However, the author did not write specifically about the program steps of the whole survey, such as the criteria to be considered for question design, etc., and it is recommended that the author make additions.
2. In line 117 of the article, the author mentions that the other survey questions were categorized into four categories, but the author later writes only three categories of sociodemographic , ownership characteristics and experience with feral swine, so if this is a clerical error, please ask the author to make corrections.
3. In Table 3, the data is incorrectly aligned, and the percentage figures in each row are not aligned with the corresponding FS control/manage options, so please ask the authors to check and correct them.
4. In the introduction, the authors suggested that feral hogs caused different damages to three different land types, and that the attitudes of land privatization towards feral hogs in the different land types were not reflected in the results. If the authors had considered and included this factor in their experimental design, it is hoped that the authors would have added to the results whether or not the attitudes of land privatizers toward feral pigs differed among the different land types in the three regions.
5. In Outcome 3.2, the authors only describe whether or not private landowners have found wild boars on their land. In order to be able to better manage wild boars and develop conservation strategies, it is recommended that the authors add a survey to find out whether or not wild boars have caused any economic losses to private landowners, and, if so, in what kind of behaviors (e.g., damage to crops).
Discussion:
1. In Result 3.2, the authors found that private landowners in the ETX region found greater ranges of feral hogs compared to the other two regions, but the authors did not discuss this difference in their discussion. It is hoped that the authors will discuss why private landowners in the ETX region found greater ranges of feral hogs around the differences between the three regions.
2. In their discussion of private landowner perceptions of feral hog control measures, the authors do not go into more depth on why measures such as trapping and killing are preferred. It is recommended that the authors further discuss the potential reasons behind private landowners' support for lethal control measures and what this may suggest for wildlife management policy and practice.
3. In the discussion, the author appears to have cited only three pieces of literature, which may not provide effective persuasion, and it is recommended that the author broaden the literature search by analyzing and discussing in depth in the discussion section how the findings of the cited literature relate to their own findings, including any similarities, differences, and possible reasons for them.
Conclusions:
In the conclusion, the authors have partially summarized the overall results of the article, however, the language may not be concise enough and it is recommended that the authors review and refine the language in the conclusion section to ensure that it clearly and accurately conveys the main findings and implications of the study. Additionally, it is recommended that the authors more clearly summarize the specific contributions of this study to the existing literature in the conclusion.
Author Response
Please see the attached response letter, thanks!

Reviewer 3 Report
Comments and Suggestions for Authors
I found the Introduction section of your paper to be quite compelling. However, considering the extensive literature available in this field, I believe it's essential to formulate hypotheses in addition to your research questions. These hypotheses would necessitate an analytical approach, which seems lacking in the current study. Allow me to elaborate on this further.
Regarding the Methods section, the presentation of your questionnaire requires improvement. I suggest two possible approaches: either include the entire questionnaire as an Appendix, or when describing the questionnaire items in the Methods section, reference tables within the Results section to indicate their presence (if not, then the first approach should be considered).
In the Results section, I noticed redundancy with the word "However." Please remove one instance for clarity. Additionally, the excessive use of acronyms can hinder comprehension. I strongly recommend using the full term "feral swine" instead of abbreviations like "FS." Also, the use of "WRG" at the beginning of the results seems unconventional and could be clarified.
While the results provide a descriptive overview, I believe they represent just the starting point of your analysis. For instance, the use of PCA and the selection of three domains could be further explored using MANCOVA or multiple regression analysis. I propose investigating the following questions:
- Does the age of participants correlate with support for lethal control (as depicted in Table 3) and/or with mean scores from the three domains presented in Table 4?
- Are there gender differences in support for lethal control (Table 3) and/or in mean scores from the three domains presented in Table 4?
- Does the income of participants influence support for lethal control (Table 3) and/or mean scores from the three domains presented in Table 4?
- To what extent does the presence of feral swine on participants' land influence support for lethal control (Table 3) and/or mean scores from the three domains presented in Table 4?
- Does the size of owned land influence support for lethal control (Table 3) and/or mean scores from the three domains presented in Table 4?
- Does the level of education influence support for lethal control (Table 3) and/or mean scores from the three domains presented in Table 4?
Furthermore, did you include a question about the amount of money stakeholders spend on damages caused by feral swine? Exploring whether individuals who incur higher expenses are more likely to support lethal control could provide valuable insights.
Finally, it would be beneficial to briefly introduce the predictors of these analyses in the Introduction section. Why might factors such as gender, education, age, or income be associated with attitudes toward feral swine? Drawing upon existing literature on this topic would strengthen the rationale behind these inquiries.
I will read your discussion after these issues will be resolved.
Overall, your paper has significant potential, and addressing these points will enhance its contribution to the field.
Author Response

(The authors gave the same response as above.)

Reviewer 4 Report
Comments and Suggestions for Authors
References should be re-arranged by MDPI requirements. I suggest to add more research e.g., in Europe and other countries.

Author Response

(The authors gave the same response as above.)

Round 2
Reviewer 1 Report
Comments and Suggestions for Authors
Good work, I have not got additional commnets.
Author Response
Thanks again for your review and comments.
Reviewer 2 Report
Comments and Suggestions for Authors
The revised version of the article has made changes to most of the previously mentioned issues, and has responded reasonably well to some of the important issues I mentioned earlier. The overall structure of the article is now more complete, however, I believe that there are still some areas of improvement in the writing of the article, and I suggest that the author may consider further optimizing the content of the article, as suggested below:
1. In the fourth paragraph of the preface of this revised version, the author is introducing the fact that wild boars may cause some impacts on the ecosystem. I suggest that the author can exchange the content of this paragraph with the third paragraph in order to make the preface as a whole present a recursive and logical relationship.
2. The authors have added a section on the generalization of the study area in the revised draft. I suggest that the authors can briefly mention the land types, climatic conditions, the number of private landowners and other relevant information of the three areas in this section, so that the readers will be able to have a clearer perception of the situation in the study area.
Author Response
Please see the attached letter for responses, thanks again!

Reviewer 3 Report
Comments and Suggestions for Authors
Perhaps if you are planning to publish another study linked with this one, the current study can has in the title "Part I" and the second study "Part II". Example: Small, E. (2011). The new Noah's Ark: beautiful and useful species only. Part 1. Biodiversity conservation issues and priorities. Biodiversity, 12(4), 232-247.
Small, E. (2012). The new Noah's Ark: beautiful and useful species only. Part 2. The chosen species. Biodiversity, 13(1), 37-53.
Author Response
Thanks for your suggestion, which is good and smart. As requested, we treat this one as Part I.